# Gaze-based Command Activation Technique Robust Against Unintentional Activation using Dwell-then-Gesture

Toshiya Isomoto*
University of Tsukuba

Shota Yamanaka†
Yahoo Japan Corporation

Buntarou Shizuki‡
University of Tsukuba

**ABSTRACT**

We demonstrate a gaze-based command activation technique that is robust against unintentional command activations using a series of dwelling on a target and performing a specific gesture (dwell-then-gesture manipulation). The gesture adopted is a simple two-level stroke, which consists of a sequence of two orthogonal strokes. To achieve robustness against unintentional command activations, we designed and fine-tuned a gesture detection system based on how users move their gaze, as revealed through three experiments. Although our technique seems to simply combine well-known dwell- and gesture-based manipulations, implying a low rate of success, our technique is actually the first technique that consists of a short time dwelling for target selection and a simple gesture for command activation. In addition, our technique will be the first technique adopting a marking menu, which is a traditional menu for command activation used in mouse- or pen-based interactions to gaze-based interactions.

**Index Terms:** Human-centered computing—Human computer interaction (HCI); Human-centered computing—Human computer interaction (HCI)—User studies; Human-centered computing—Human computer interaction (HCI)—Gestural input

## 1 INTRODUCTION

In a gaze-based interaction, preventing a user's unintentional manipulation is a challenge because it is difficult to distinguish whether a gaze movement is intended for activating a command or not. Because a gaze movement occurs in every human activity such as reading a sentence or watching a movie, addressing this challenge (i.e., preventing user's unintentional manipulation) is crucial for a gaze-based interaction as compared with other mouse- or pen-based interactions, for example. For this reason, significant effort has gone into making a gaze-based interaction robust against unintentional manipulations.

For example, to prevent unintentional dwell-based target selections (a problem referred to as a Midas-touch [14]), researchers have explored techniques that work even when the dwell time (the time during which users look at an object to select it) is short (e.g., [2, 12, 24]). With these techniques, intentional dwelling is identified by detecting such dwelling on a specific object (e.g., [24]), from the users' next action (e.g., [2]), or from the users' gaze movement (e.g., [12]). Similarly, in gesture-based manipulation, which uses specific gaze movements (gestures) for command activation, a command is activated when users perform a gesture defined beforehand and the gesture is detected by a system. However, in such cases, a complex gaze movement (e.g., found in [4, 5]) is used to allow the system to clearly distinguish whether a gaze movement is intentional.

---

*e-mail: isomoto@iplab.cs.tsukuba.ac.jp
†e-mail: syamanak@yahoo-corp.jp
‡e-mail: shizuki@cs.tsukuba.ac.jp

Graphics Interface Conference 2020
28-29 May

Among the two types of manipulations described above, gesture-based manipulations are considered more suitable for command activation in terms of the command activation time and robustness against unintentional manipulations [6, 11]. In addition, techniques that combine dwelling and gestures have been proposed [4, 27]; a dwell-based target selection has been used to select an object that a user wants to activate a command (e.g., one icon in a crowd of icons), which also displays visual guidance (e.g., a menu), and the command is then activated through a gesture. Traditionally, such a technique has been used for mouse-based manipulations; when a user right-clicks on an object, the object is temporarily selected, and visual guidance is displayed. Although this scheme works with mouse-based manipulation, in a gaze-based interaction, it currently does not work because a gaze-based interaction still faces the problem of unintentional manipulations. For example, an unintentional target selection causes an unintentional display of visual guidance, and an unintentional gesture detection causes an unintentional command activation.

The combination of dwell- and gesture-based manipulation has significant potential for use in a gaze-based command activation; however, the negative effects of each manipulation (i.e., the Midas-touch problem or complexity of the gesture) remain. We assume that this is because previous researchers simply connected dwell-based manipulation with gesture-based manipulation. Therefore, herein, we properly combine both manipulations as a *series* of dwell-based and gesture-based (dwell-then-gesture) manipulations and remove their negative effects. The adopted gesture is a simple two-level stroke, which consists of a sequence of two orthogonal strokes. A gesture detection system is designed based on how users move their gaze, as revealed through three experiments. In the first experiment, we characterize the gaze movements and determine the parameters used for the detection system. Through the following two experiments, we fine-tune the detection system to achieve more robustness against an unintentional manipulation without losing the easiness of the manipulation. This system detects a user's *intentional* series of dwell-then-gestures. With our dwell-then-gesture technique, users can activate a command, as shown in Figure 1.

## 2 RELATED WORK

For each dwell- and gesture-based manipulation, methods for command activation have been proposed. We first state why a gesture-based command activation is more attractive than a dwell-based activation by describing the problem of the latter. We then clarify the necessity of a careful design of gaze-based gesture-based manipulation.

### 2.1 Dwell-based Command Activation

Although activating a command using dwell-based manipulation is simple and therefore easy-to-learn (i.e., a user simply looks at an object (e.g., an icon or menu) that the command is associated with or that the user wants to activate the command on), it faces a Midas-touch problem. Some researchers have sought to alleviate this problem with a short dwell time [2, 12, 24]; however, recent dwell-based manipulations still face this problem. Using a GUI such as a pull-down or pie menu [3], the user needs to dwell on an object twice or more to select the command from the GUI. For example,

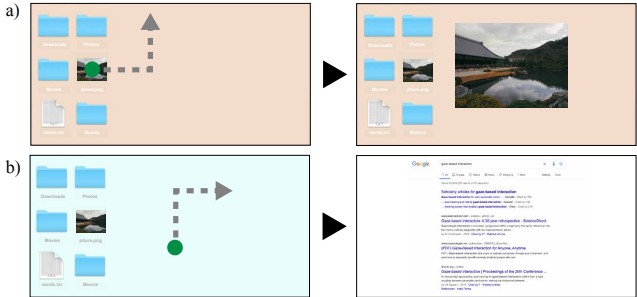

Figure 1: Applications of our technique. A user first dwells on a point where the user wants to activate a command. Then, performing a) a right-then-up stroke opens the icon if the pointer is on an icon ("opening an image file" in this case), or performing b) an up-then-right stroke on a window or the desktop switches the foreground window ("switching from a desktop window to a browser window" in this case).

the user first dwells on a menu to display its items; the user then dwells on one of the menu items to activate the command. With this scheme, a Midas-touch causes an unintentional object selection and the menu to be frequently displayed.

Another technique for preventing an unintentional manipulation is pursuit-based target selection in which a *smooth pursuit* is detected [7, 28, 29]. However, Špakov et al. [30] showed that a dwell-based technique performs as equally as or better than two pursuit techniques (where a smooth pursuit is caused by an object moving in a circle or linearly). Using a pursuit-based target selection twice, a command activation can be performed comparable to a dwell-based manipulation.

Researchers have examined the performance of a dwell-based command activation and compared it with a gesture-based command activation. The experimental results showed that a command activation with linear and ring-shaped hierarchical menus requires over 4.0 s simply for selecting a menu item, whereas the success rates for a two-level depth were over 90% (false-positives were not counted) [17]. The comparison results between gesture- and dwell-based manipulation [6, 11] show that gesture-based manipulation is faster and achieves lower error rates than dwell-based manipulation. Based on these results, gesture-based manipulation has been established as a command activation technique.

## 2.2 Gesture-based Command Activation

In gesture-based command activation, a user can activate a command by performing a gesture defined beforehand. Herein, we categorize them into two types: a one-level stroke, which is a gesture such as a right-to-left gaze movement [21–23, 25] and a multi-level stroke, which is a combination of two or more one-level strokes [5, 13, 32]. In terms of robustness against unintentional manipulation, a multi-level stroke is superior to a one-level stroke; however, activating a command with a multi-level stroke is more difficult owing to its complexity. Therefore, with certain techniques, visual guidance (e.g., a menu) is adopted to help users to easily activate a command; for example, an additionally displayed window [32], a semi-transparent region [13], or a physical object [16]. However, compared with other manipulation methods such as using a mouse, users cannot select an object on which to activate a command.

With certain techniques, a combination of dwell-based (or fixation-based) target selection and gesture-based command activation is used; the target selection also displays visual guidance. For example, using a pie menu, the menu is displayed after fixation, and the command is activated when the gaze crosses the edges of the menu [10, 27]. The principle of a "screen button" [31] is similar to this combined manipulation. To select an object, users need to look

at the object, move their gaze to a constantly displayed button, and fixate on it (i.e., a combination of fixation and a one-level stroke). Although these combined techniques show the potential for gaze-based command activations, a fixation may lead to unintentional manipulations, because a fixation, that is, a dwell performed within a short dwell time, could cause more Midas-touch problems, which may cause an unintentional display of a pie menu, or an unintentional command activation owing to unintentional one-level stroke detection. As a reasonable solution, Delamare et al. [4] adopted pursuit-based gestures to a command activation because pursuit is used as a gesture that is robust against an unintentional manipulation and is easy to manipulate. However, activating a command with a pursuit-based gesture requires visual guidance; thus, a long dwell time such as 2 s, is used to prevent an unintentional display (actually, head-rotation-based dwelling, which also faces the occurrence of a Midas-touch problem, is used for the AR context [4]). In addition, a 1 s of dwell time is additionally needed to hide the guidance. Note that an object selection is an easy-to-use and robust function in other manipulation methods such as a mouse and touchscreen; by contrast, in a gaze-based interaction, because of the difficulty in distinguishing unintentional gaze movements from intentional ones, even a fundamental function such as an object selection still suffers from unintentional manipulation.

In contrast to a gesture detection system used in prior techniques, in which the negative effects of each dwell- and gesture-based manipulation remain, we designed a gesture detection system that detects a user's intentional series of dwell-then-gesture manipulations. Using our technique, a user can activate a command by performing a dwell and then a simple two-level stroke.

## 2.3 Relevance of Our Work

Our dwell-then-gesture command activation technique is the first technique that consists of a short time dwelling for target selection and a simple gesture for command activation. Although prior techniques that combine dwelling and gestures have advantages in terms of no longer needing to require users to memorize a command and visually helping users perform complex gestures (e.g., pursuit-based manipulation is based on a moving visual object), users cannot activate a command without visual guidance. By contrast, our dwell-then-gesture technique can activate a command without visual guidance thorough a simple gesture. Moreover, our technique has the potential in that users can display a visual guidance if they need it; thus, our technique will become the first technique adopting a marking menu, which is a traditional menu for command activation, such as in a mouse- or pen-based interaction, to a gaze-based interaction.

## 3 EXPERIMENT 1

The goal of Experiment 1 was to find the characteristics of gaze trajectories such as the variability in horizontal, vertical, and diagonal movements when users move their gaze. The characteristics of the gaze movement is intended for determining tunable parameters, which we will use in the design of a gesture detection system.

## 3.1 Apparatus

We used a Tobii EyeX as an eye tracker. A 24-inch display with a non-glare property for preventing reflections with a resolution of 1,920 × 1,080 pixels was applied. The participant's head was positioned approximately 60 cm from the display. We used a chin rest to prevent any movement of the participant's head from affecting the performance of the eye tracker. The display was placed in front of a white wall to prevent interference from objects unrelated to the experiment. Experiment 1 was conducted in a room lit by fluorescent lamps to ensure consistent lighting conditions.

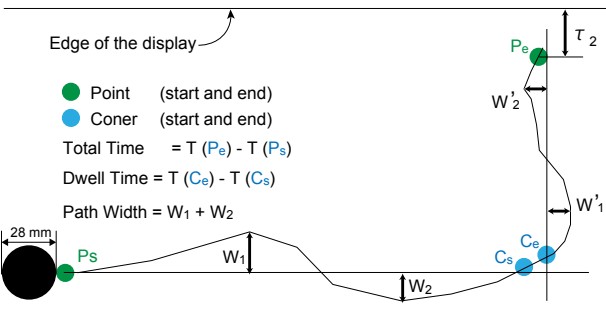

Figure 2: Analysis used in Experiment 1.

## 3.2 Participants

We recruited 16 participants (14 male, 2 female) aged 20 to 24 years (Mean = 22.0). The participants had normal or corrected (glasses or contact lenses) vision with no color vision abnormalities; six wore glasses and two wore contact lenses. Eleven had previously participated in an experiment using an eye tracker.

## 3.3 Tasks and Procedure

To obtain gaze trajectories when users move their gaze, we asked the participants to conduct the following tasks:

**one-level task** The participants were asked to move their gaze from the center of the display through one of four gestures (to the upper right, *UR* ↗; upper left, *UL* ↖; lower right, *DR* ↘; or lower left, *DL* ↙).

**two-level task** The participants were asked to move their gaze from the center of the display through one of eight gestures, each consisting of a sequence of two orthogonal strokes ($U{\rightarrow}R$ ↱, $U{\rightarrow}L$ ↰, $D{\rightarrow}R$ ↳, $D{\rightarrow}L$ ↲, $R{\rightarrow}U$ ↱, $R{\rightarrow}D$ ↴, $L{\rightarrow}U$ ↰, or $L{\rightarrow}D$ ↴). They were also told that there was no need to dwell intentionally when they changed their gaze movement from the first direction to the second direction.

To ensure a consistent starting gaze point, the participants were asked to look at a black circle (28 mm in diameter) shown at the center of the display and to then move their gaze. The trial began after the participants pressed a key on the keyboard placed on the desk, and a recorded voice-based instruction such as "look to the upper right" for a one-level task or "look upward, then right" for a two-level task was then played. We also asked the participants to re-press a key after they finished the trial. To reduce the effects of fatigue, the participants were offered optional breaks between trials as well as a mandatory break of more than five minutes after conducting 32 trials. To counterbalance the effects of the order, eight participants started with the one-level task, whereas the others started with the two-level task. The order of the instructions of the gestures was randomized for every four trials (gestures) under one-level task and for every eight trials (gestures) under two-level task. Before each task, the participants calibrated the eye tracker. Note that we did not provide the participants with feedback regarding the gaze position to avoid confusion from the offset of the position detected by the eye tracker [15].

Each participant conducted 16 trials in each direction. In total, we collected 1,024 (64 trials × 16 participants) trials for one-level task and 2,048 (128 trials × 16 participants) trials for two-level task. This experiment took approximately 52 minutes per participant, and each participant was paid 5,000 JPY (approximately 45 USD).

## 3.4 Analysis of Gaze Trajectories

We first excluded the trials in which the gaze moved against the instructions with the angle-based processing. For example, when the

instruction is to conduct a $U{\rightarrow}R$, a gaze movement that goes into Quadrants II-IV is an outlier (3.7% in the one-level task); when the instruction is $U{\rightarrow}R$, the first level movement that goes downward, or the second level movement that goes left or downward is an outlier (9.8% in the two-level task). Next, we plotted the remaining trials and manually excluded those in which directions and gaze movements were correct however, obviously against the task (e.g., gazes moving orthogonally during a one-level task or diagonally during a two-level task); 1.9% and 2.0% were shown to be outliers for the one- and two-level tasks, respectively. The simplicity of the task causes a human error and such outliers. In fact, we frequently heard that the participants comment that they made a mistake. As a result, 5.6% of the trials for a one-level task and 11.8% of the trials for a two-level task were excluded. That is, we treated 94.4% of the one-level tasks and 88.2% of the two-level tasks as successful trials.

The gaze trajectories of the successful trials for the two-level task had two characteristics. In one, the gaze stayed at the corner between the first- and second-level movements. In the other, the gaze did not move in a perfectly horizontal or vertical direction; rather, it moved slightly diagonally or in a slightly zigzag manner.

To quantify the characteristics, we applied the process below. First, we applied a low-pass filter (i.e., $P_i = 0.25p_i + 0.75P_{i-1}$, where $P_i$ is the $i$-th low-pass filtered gaze point; $p_i$ is the $i$-th raw gaze point sampled from the eye tracker) to the data of successful trials to eliminate the dispersion of the gaze. Next, we identified four points in the gaze trajectory for each trial, as shown in Figure 2: the point where the gaze starts moving ($P_s$), the point where the gaze starts cornering ($C_s$), the point where the gaze starts moving from $C_s$ ($C_e$), and the point at which the trial is finished ($P_e$). The four points were identified as follows:

**$P_s$** This is the point where $d_x$ (the differential of coordinate $x$ from the previous sample) was more than the threshold $\tau_1$, and the gaze point was outside the black circle when the first-level stroke was toward the right. When the first-level stroke was upward, we used $d_y$ (the differential of coordinate $y$ from the previous sample) and $-\tau_1$ instead of $d_x$ and $\tau_1$, respectively.

**$C_s$** This is the point where $d_x$ was less than $\tau_1$ after $P_s$ was identified. When $d_x$ was more than $\tau_1$, the identification of $C_s$ was repeated.

**$C_e$** This is the point where $d_y$ was more than $\tau_1$ away from $C_s$ and larger than $d_x$.

**$P_e$** This is the point where both the $x$ and $y$ coordinates of the gaze point moved less than another threshold $\tau_2$ from the top-right corner of the display.

During this process, 5 mm was adopted as $\tau_1$, which is the accuracy of the eye tracker obtained from the experimental results in [8]; that is, the gaze was recognized with an offset of 0–5 mm with Tobii EyeX applied in this experimental environment. Moreover, we determined $\tau_2$ to be 55.5 mm, so that we could find $P_e$ for all successful trials of Experiment 1.

## 3.5 Characteristics of Gaze Trajectories

We derived the time during which the gaze stayed at the corner by calculating the difference between the timestamps of $C_s$ and $C_e$. The average time was 133 ms (SD = 86) for the successful trials of two-level task (note that there was no corner for one-level task).

We derived the width of the gaze trajectories in the successful trials from how orthogonally the gaze moved against the direction of the instruction. For example, in Figure 2, the direction of the first-level stroke is horizontal; thus, the width is the distance between the highest and lowest $y$ coordinates of the gaze between $P_s$ and $C_s$ (i.e., $W_1 + W_2$) In Figure 2, the direction of the second-level stroke was vertical; thus, the width of trajectory is the sum of $W'_1$ and $W'_2$. The average widths of the trajectories in the first- and second-level stroke were 17.0 mm (SD = 25.9) and 17.8 mm (SD = 24.1), respectively.

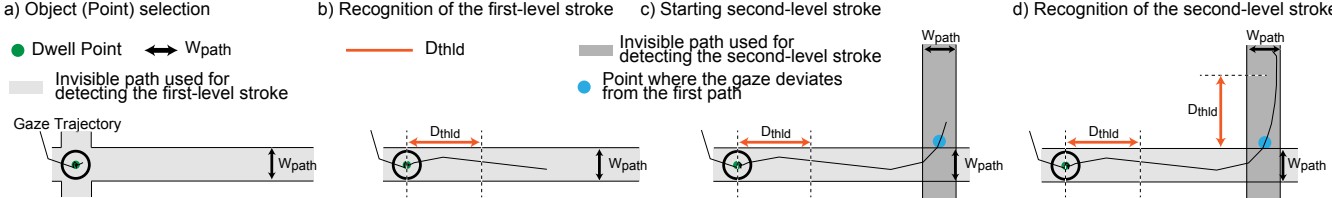

Figure 3: Method of detecting two-level stroke.

## 4 GESTURE DETECTION SYSTEM

We used the gaze trajectories obtained during Experiment 1 to design the gesture detection system for detecting intentional two-level strokes.

### 4.1 Overview of System

Our detection system works as follows (Figure 3).

a) Once the point/object, on which a command should be activated, is selected by dwelling, an invisible straight path for detecting the first-level stroke is generated, with the center at the dwell point and width $W_{path}$. With our technique, a dwell is detected when the gaze stays within 5 mm ($\tau_1$) for the dwell time ($T_{dwell}$).

b) The first-level stroke is detected when the gaze moves horizontally/vertically along a path longer than the threshold $D_{thld}$ from the dwell point.

c) The path for detecting the second-level stroke is generated at the point $P$ where the gaze deviates from the first path, with the center at $P$ and the width $W_{path}$.

d) The second-level stroke is recognized when the gaze moves vertically/horizontally along the path longer than $D_{thld}$ from $P$. After detecting the second-level stroke, the command is activated on the point/object.

The $T_{dwell}$, $W_{path}$, and $D_{thld}$ should be neither too large nor too small, meaning that they should be empirically determined. Specifically, $W_{path}$ and $D_{thld}$ contribute to preventing unintentional command activations in this detection system. If $W_{path}$ is extremely large and $D_{thld}$ is extremely small, the number of the unintentional command activations becomes large; otherwise, activating a command becomes intentionally difficult. Moreover, the detection of a two-level stroke and $T_{dwell}$ strongly influence each other. If the system can avoid unintentional two-level stroke detections, we can use a small $T_{dwell}$ without considering a Midas-touch.

### 4.2 Determination of Parameters

We determined $T_{dwell}$, $W_{path}$, and $D_{thld}$ based on the characteristics found in Experiment 1.

#### 4.2.1 $T_{dwell}$

In general, an overly long $T_{dwell}$ decreases the usability of the dwell-based target selection. Moreover, an extremely short $T_{dwell}$ can not be used because we found that the gaze stayed at the corner between the first- and second-level strokes. We thus determined the $T_{dwell}$ as the sum of the average and SD at the time the gaze stayed at the corner. As a result, the $T_{dwell}$ was 219 ms (133+86), which is within the 85.9th percentile of all successful trials of the two-level task.

#### 4.2.2 $W_{path}$

To prevent an unintentional command activation, $W_{path}$ must be small; however, to make it easier to perform a command activation, $W_{path}$ must be sufficiently large because a small $W_{path}$ makes activating a command difficult in that users need to move their gaze in straight lines that are orthogonal to each other. We determined

$W_{path}$ as the sum of the average and SD of the widths of the gaze trajectories. The sums were 42.9 mm (17.0 + 25.9) and 41.9 mm (17.8 + 24.1) for the first- and second-level strokes, respectively; a Wilcoxon signed-rank test showed no significant difference (p > 0.05). Therefore, we adopted 42.9 mm as $W_{path}$, which is within the 93.2th percentile of all successful trials of two-level task.

#### 4.2.3 $D_{thld}$

We attempted to determine $D_{thld}$ from the results; however, the gaze tended to move to the edge of the display in the first- and second-level strokes (e.g., when the instruction was $R{\rightarrow}U$, the gaze approached the right edge and then approached the upper edge of the display), and hence, we could not determine $D_{thld}$ from the results.

Instead, we determined $D_{thld}$ by testing various $D_{thld}$ values (27.9 mm, 41.9 mm (1.5 × 27.9 mm), and 55.8 mm (2.0 × 27.9 mm)); we simulated how many gaze trajectories of one-level task and two-level task could be recognized with each $D_{thld}$. When $D_{thld}$ was 27.9 mm, the recognition rates were 9.9% and 74.5% in the one- and two-level tasks, respectively. When $D_{thld}$ was 41.9 mm, the recognition rates were 0.5% and 15.8%. When $D_{thld}$ was 55.8 mm, the recognition rates were 0.0% and 5.4%. Note that the recognition rate for the one-level task indicates the rate of unintentional command activation, whereas the recognition rate for the two-level task indicates the rate of intentional command activation. From this result, we determined $D_{thld}$ to be 55.8 mm with the intention of preventing an unintentional command activation and under the assumption that the recognition rate for the two-level task might improve as the users become familiar with our technique.

## 5 EXPERIMENT 2

The goals of Experiment 2 were to fine-tune our system to improve its performance and determine whether it can be used by starting from the corners of the display. These are because we developed our system by focusing on an unintentional manipulation and developed it using gaze trajectories starting from the center of the display, respectively. To investigate how the performance improves through optimization, we first obtained the gaze trajectories when the participants tried to activate a command using our technique and then simulated the performance using the trajectories. The apparatus and experimental environment were the same as in Experiment 1. We also adopted the same low-pass filter used in the analysis of Experiment 1.

### 5.1 Participants

We recruited 16 participants (all males) aged 21 to 24 years (Mean = 23.0). The participants had normal or corrected vision with no color vision abnormalities. Two of the participants wore glasses. Eight of them had participated in Experiment 1; we referred to these participants as Group A, and the other eight as Group B. Thirteen had previously participated in an experiment using an eye tracker.

### 5.2 Tasks and Procedure

Figure 4 shows the display used in Experiment 2. A target was located at one of five positions: top-right (TR), top-left (TL), bottom-right (BR), bottom-left (BL), and center (C). The eight gestures of

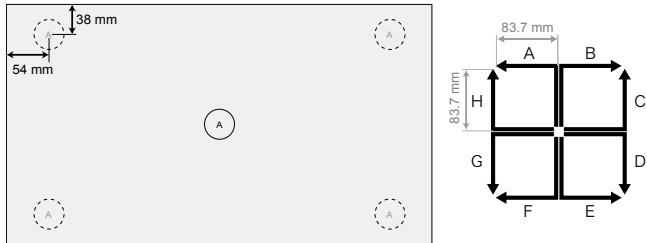

Figure 4: Display used in Experiment 2. The dotted circles in the left figure indicate the positions of the targets. The circles indicate the target, and the letter in the circle ('A' in this case) corresponds to a gesture of the cheat sheet (right figure).

the two-level strokes were mapped to the letters shown in Figure 4 right. When the participants dwelled on a target, the letter turned red.

A trial consisted of dwelling on a target for 219 ms ($T_{dwell}$) and then performing a two-level stroke corresponding to the displayed letter. Experiment 2 had three display phases (rest phase, performance phase, and instruction phase); the phases were switched in sequence by pressing the 'Enter' key placed on the desk. The experiment began with rest phase in which no target was displayed. In rest phase, the participants could take a break at their leisure. They pressed the 'Enter' key when they finished resting. In performance phase, a target and a letter were displayed. The participants could check the cheat sheet (Figure 4, left) by pressinf the 'Space' key during performance phase. We also asked the participants to press the 'Enter' key when they finished the trial. After pressing the 'Enter' key, the phase became instruction phase, and the participants were notified of the detection result (success or failure) by a sound. In instruction phase, the trajectory of the gaze during the two-level stroke was displayed.

A session consisted of 40 (8 gestures × 5 positions) trials; six sessions were conducted. The first three sessions were used for practice, whereas the other three sessions were the actual test. In instruction phase of the practice sessions, to increase the participants' familiarity with our technique, an experimenter told the participants their results and the reasons for their failures. For example, the participants were told that the detection failed because the gaze moved diagonally. By contrast, the experimenter did not describe the results and the reasons during the test sessions. Before the first session, the participants calibrated the eye tracker. In total, we collected 1,920 (40 trials × 3 sessions × 16 participants) trials of data from both the practice and test sessions. We did not provide feedback regarding the gaze position for the same reason as in Experiment 1. After finishing all sessions, the participants were asked to complete a questionnaire about our technique and asked to take at least a five minutes of rest. This experiment took approximately 85 minutes per participant; each participant was paid 5,000 JPY (approximately 45 USD).

## 5.3  Discussion of Experimental Results

Below, we discuss the success rate and activation time in the center and corners of the display and at other positions. Next, we discuss the reasons for the results. The success rate is the sum of successful trials (trials in which the recognized and instructed gestures were the same ) divided by the total number of trials. The activation time was from the start of dwelling on a target to the time when the two-level stroke was detected.

In the test sessions, the average success rates were 77.6%, 50.8%, 61.7%, 70.1%, and 67.5% at C, TL, TR, BL, and BR, respectively. The activation times were 813 ms, 894 ms, 895 ms, 815 ms, and 826 ms at C, TL, TR, BL, and BR, respectively. The rates were not

as high at all positions. As the reasons for this result, we found that the 219 ms of $T_{dwell}$ was too small for two reasons. First, during the practice sessions, the participants' gaze frequently stayed for a considerably long time at a corner, and we informed them of this problem. Second, four of the participants answered that it was difficult to not stop gazing at a corner, whereas we received no comments regarding the $W_{path}$ or $D_{thld}$. Note that a larger $T_{dwell}$ may cause more unintentional manipulations. Therefore, we simulated the success rate with longer $T_{dwell}$s.

With respect to the success rate at the corners of the display, a Wilcoxon signed-rank test showed no significant difference between C and BR ($p > 0.05$). As the reason for this result, five of the participants answered that they felt difficulty in moving their gaze outside the display. Moreover, 11 participants answered that it was easy to move their gaze from the center to the corner and from the corner to the center. The average success rate in this case (i.e., when the combinations of the position and gesture were TR and $L{\rightarrow}D$ or $D{\rightarrow}L$, TL and $R{\rightarrow}D$ or $D{\rightarrow}R$, BR and $L{\rightarrow}U$ or $U{\rightarrow}L$, and BL and $R{\rightarrow}U$ or $U{\rightarrow}R$) was 71.4%. These results show that the detection system developed in Experiment 1 cannot be used at the corners of the display and indicate the necessity of tuning the parameters depending on the location of the contents (i.e., starting location of the gesture).

## 5.4  Simulation Results

To investigate the effect of various $T_{dwell}$s during the test session, we simulated the success rate at C every 10 ms, from 219 ms of $T_{dwell}$ to 1000 ms. The result of this simulation showed that the success rate increased as the $T_{dwell}$ became large and reached the highest rate of 88.0% when the $T_{dwell}$ was 490 ms (remaining at 88.0% even after the $T_{dwell}$ became larger than 490 ms). We also derived the highest success rate for $T_{dwell}$s of 219 ms, 306 ms, 392 ms, and 478 ms; these $T_{dwell}$s were calculated by the sum of the average and one, two, three, and four SDs of the time during which the gaze stayed at the corner in Experiment 1, respectively. The success rates for these four $T_{dwell}$s were 77.6%, 84.9%, 87.0%, and 87.5%. Wilcoxon signed-rank tests showed no significant differences between 306–392 ms and 392–478 ms ($p > 0.05$). The success rates with respect to the gestures were 75.0% in $R{\rightarrow}U$, 87.5% in $R{\rightarrow}D$, 85.4% in $U{\rightarrow}R$, 97.9% in $U{\rightarrow}L$, 70.8% in $L{\rightarrow}U$, 75.0% in $L{\rightarrow}D$, in 93.8% in $D{\rightarrow}R$, and 93.8% in $D{\rightarrow}L$. The activation times for the four $T_{dwell}$s were 813 ms 912 ms, 1007 ms, and 1096 ms. Regarding the success rate and activation time, Mann-Whitney tests showed no significant difference between Groups A and B. These results suggest that using 306 ms as the $T_{dwell}$ is suitable for both the success rate and robustness against unintentional manipulation.

## 6  Experiment 3

The goals of Experiment 3 were to determine how to fine-tune the gesture detection system to make it more robust against unintentional dwell-then-gesture detections and to investigate the participants' impressions of it. We adopted 306 ms as the $T_{dwell}$, 42.9 mm as the $W_{path}$, and 55.8 mm as the $D_{thld}$. We used a Tobii Eye Tracker 4C and did not use a chin rest. The other apparatuses and experimental environment were the same as in Experiments 1 and 2.

## 6.1  Participants

We recruited 16 participants (all male, Japanese) aged 21–25 years (Mean = 22.5). The participants had normal or corrected vision with no color vision abnormalities; three wore glasses, and two wore contact lenses. Nine had previously participated in an experiment using an eye tracker, and two had used an eye tracker for gaming (none had used one for any other activities). Eight participated in Experiment 2 (Group C); the others had not (Group D).

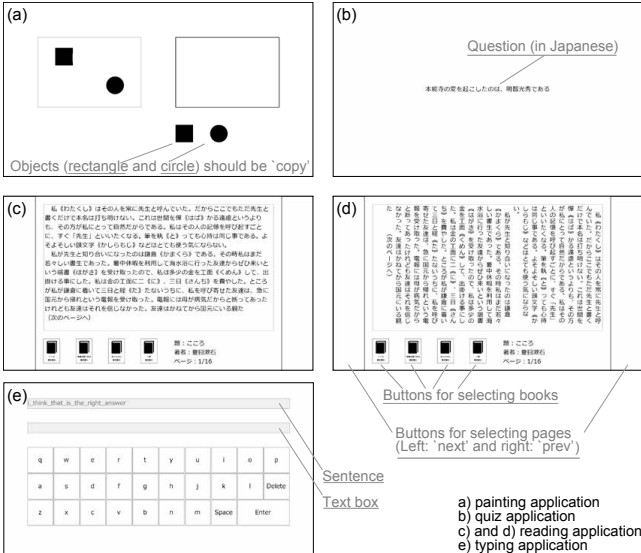

Figure 5: Applications used in Experiment 3: a) painting, b) quiz, c) and d) reading, and e) typing.

## 6.2 Applications in Experiment 3

Figure 5 shows the applications used in the experiment: a) painting, b) quiz, c) reading of horizontally presented sentences, d) a reading of vertically presented sentences, and e) typing. The painting and quiz applications were manipulated using our technique; the others were manipulated with a 1 s dwell-based selection. In the painting application, the participants were asked to arrange the objects (a rectangle and a circle) at roughly the same positions as in an image displayed on the left side of the screen by activating a 'copy' and 'paste' twice (once for the rectangle object and once for the circle object). In the quiz application, the participants were asked to answer four questions by activating 'yes' or 'no' using our technique. In the reading application, the participants were asked to read an e-book by manipulating the application; they could choose a book from among four books and turn their pages. Because all participants were Japanese, the sentences used in the quiz and reading application were in Japanese. In the typing application, the participants were asked to type sentences consisting of 32 characters including spaces. These sentences were extracted from [20]. We refer to the painting and quiz applications as gesture-driven applications and the reading and typing applications as dwell-driven applications.

The purposes of the dwell-driven applications were to promote horizontal gaze movements in the horizontal reading task, to promote vertical gaze movements in the vertical reading task, and to promote dwelling and horizontal/vertical gaze movements during the typing task.

## 6.3 Tasks and Procedure

The participants were asked to use the painting application twice and then quiz application twice. They were also asked to read e-books with the vertical and horizontal reading applications in that order for five minutes each and then type five sentences with the typing application. To investigate whether familiarity with our technique affected the number of unintentional detections, we asked the participants of Group C to begin with the gesture-driven applications and those of Group D to begin with the dwell-driven applications. The eye tracker was calibrated prior to this experiment.

The tasks using the gesture-driven applications began after the practice of the dwell-then-gestures; the practice was finished when eight correct intentional gestures of strokes were consecutively de-

Table 1: Mappings of commands and strokes used in Experiment 3.

| | command | first | second |
|---|---|---|---|
| painting | copy | $U{\rightarrow}R$ | $U{\rightarrow}L$ |
| | paste | $D{\rightarrow}L$ | $D{\rightarrow}R$ |
| quiz | yes | $R{\rightarrow}D$ | $R{\rightarrow}U$ |
| | no | $L{\rightarrow}U$ | $L{\rightarrow}D$ |

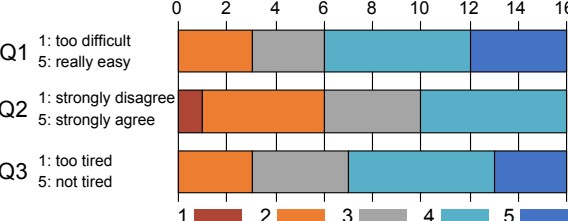

Figure 6: Answers to the questionnaire for the gesture-driven applications.

tected. The participants were then asked to perform each task twice (i.e., painting→painting→quiz→quiz); the mappings of the strokes and commands are shown in Table 1. For example, in the first painting application task, the participants could activate 'copy' an object by dwelling on a rectangle and performing a $U{\rightarrow}R$ stroke, and 'paste' the copied object by dwelling on a point at roughly the same position in the left image and then by performing a $D{\rightarrow}L$ stroke; in the second painting application, the $U{\rightarrow}L$ stroke was for 'copy' and the $D{\rightarrow}R$ stroke was for 'paste'. Each stroke for 'copy' and 'paste' was performed twice per task. In the quiz application, the participants could activate 'yes' and 'no' by dwelling on a point somewhere on the display and performing an $R{\rightarrow}D$ ($R{\rightarrow}U$) or $L{\rightarrow}U$ ($L{\rightarrow}D$) stroke, respectively. Each stroke for 'yes' and 'no' was performed twice per task. In total, we collected 256 gaze trajectories (8 gestures × 2 activations × 16 participants) as the intentional command activation.

The task using the dwell-driven applications began after an explanation of the dwell-based manipulation; we told the participants that they had to look at the button for 1 s in order to select it.

After completing each task, the participants were asked to take a break for more than five minutes. Moreover, after finishing the tasks using the gesture-driven applications and the tasks using the dwell-driven applications, the participants answered a questionnaire on a 5-point Likert scale (1 = negative and 5 = positive). The questionnaire had three questions, Q1, "Did you feel that the gaze movement was easy?"; Q2, "Did you feel that the gesture was successfully detected?"; Q3, "Did you feel tired?", and extra space for any additional comments. The average time taken for practice before the tasks using the gesture-driven applications was approximately 212 s (SD = 93). On average, the task using the dwell-driven applications took approximately 17 minutes (5 minutes of horizontal reading tasks + 5 minutes of horizontal reading tasks + 7 minutes of a typing task). In total, the experiment took approximately 55 minutes per participant; each participant was paid 5,000 JPY (approximately 45 USD).

## 6.4 Impressions of command activation using our technique

Figure 6 shows the answers to the questionnaire for the gesture-driven applications. Among the three questions, regarding Q2, six participants felt that the detection of the two-level strokes was unsuccessful. However, all participants successfully completed the gesture-driven applications. These results suggest that they had to perform the dwell-then-gesture two or more times before they succeeded in activating the commands. In accordance with the 84.9% success rate in Experiment 2, the participants should be able to activate a command with a 97.7% $(1-(1-0.849)^2)$ success rate if

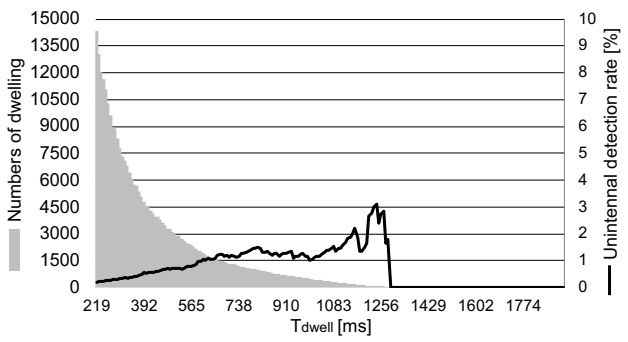

Figure 7: Number of dwell detections and unintentional detection rate against the $T_{dwell}$.

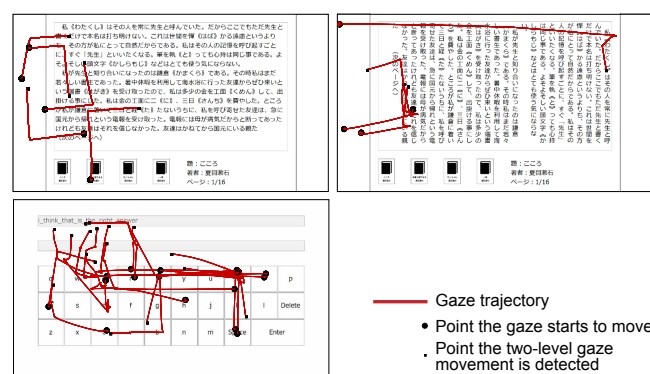

Figure 8: Trajectories of two-level strokes caused when unintentional dwell-then-gestures were detected.

they attempt the dwell-then-gesture twice. Moreover, because all participants completed all tasks, practicing for at least approximately 212 s (the time taken for practice) would be sufficient for users to become familiar with our technique.

### 6.5 Number of unintentional manipulations

The system detected 26 unintentional dwell-then-gestures (i.e., unintentional detection) during all tasks of the dwell-driven applications for a $T_{dwell}$ of 306 ms, $W_{path}$ of 42.9 mm, and $D_{thld}$ of 55.8 mm. A Mann-Whitney test showed no significant difference between Groups C and D. A total of 3 $R{\to}U$ ($\nearrow$), 3 $R{\to}D$ ($\searrow$), 3 $U{\to}R$ ($\rightarrow$), 5 $U{\to}L$ ($\leftarrow$), 10 $L{\to}U$ ($\uparrow$), 1 $L{\to}D$ ($\downarrow$), 0 $D{\to}R$ ($\rightarrow$), and 1 $D{\to}L$ ($\leftarrow$) were demonstrated. To improve our system's robustness against these unintentional detections, we simulated the number of unintentional detections by changing the parameters. In addition, to fine-tune our system, we also examined the characteristics of the gaze trajectories when an unintentional detection was occurred.

#### 6.5.1 Simulation by Changing the Parameters

We simulated the number of unintentional detections during all tasks of the dwell-driven applications for all combinations of 179 $T_{dwell}$s (from 219 ms to 2000 ms in 10 ms steps), 11 $W_{path}$s (from 0.0 mm to 42.9 mm in 4.2 mm steps), and 87 $D_{thld}$s (from 55.8 mm to 537.0 mm, i.e., the width of the display, in 5.6 mm steps). We found that there were numerous combinations for which there were no unintentional detections. Among them, to facilitate the command activation, we tried to choose the combinations having a large $T_{dwell}$, large $W_{path}$, and small $D_{thld}$ from those that had no unintentional detections.

Initially, we thought a larger $T_{dwell}$ would result in more unintentional detections. However, the simulation showed that the number of unintentional detections decreased as the $T_{dwell}$ increased, going from 26 to 0 as the $T_{dwell}$ increased from 306 ms to 1,290 ms. Figure 7 shows the number of dwell detections and unintentional detection rates (calculated by dividing the number of unintentional detections by the number of dwell detections) when we changed only the $T_{dwell}$; thus, $W_{path}$ and $D_{thld}$ were 42.9 mm and 55.8 mm, respectively. As this figure shows, the rate tended to increase as $T_{dwell}$ increased. These results show that increasing only the $T_{dwell}$ is inadequate; however, a long $T_{dwell}$ seems to work if it is combined with a fine-tuned $W_{path}$ and $D_{thld}$.

Therefore, we next investigated the effect of varying the $W_{path}$ and $D_{thld}$ when the $T_{dwell}$ was 306 ms. In this case, numerous combinations achieved zero unintentional detection; for example, there were no detections for $W_{path}$ and $D_{thld}$ combinations of (5.3 mm, 55.8 mm) or (42.9 mm, 117.0 mm). Although no unintentional detections will occur if we use a small $W_{path}$ (e.g., less than 5.3 mm) or a large $D_{thld}$ (e.g., more than 117.0 mm), the success rate may become low. Therefore, we explored the combinations to find the

largest $W_{path}$ and the smallest $D_{thld}$ combination that would still achieve zero unintentional detections; this combination turned out to be (13.7 mm and 100.0 mm). Even though this combination seemed to be optimal, $W_{path}$ was still rather small and $D_{thld}$ was large. Figure 8 shows the trajectories of 26 unintentional two-level strokes and their start point (i.e., dwelling point) during this experiment. As this figure shows, some of the trajectories were straight without any zigzags; this suggests that the $W_{path}$ may have less of an effect in preventing unintentional detections. Moreover, the horizontal trajectory (e.g., the gaze movement to the left in $L{\to}D$) seems to be longer than the vertical trajectory (e.g., the gaze movement to the upper in $L{\to}D$). Therefore, we explored the effect of using two $D_{thld}$s, one for horizontal movement ($D_h$) and one for vertical movement ($D_v$). From the combinations of $D_h$ and $D_v$ achieving zero unintentional detection, we chose the combination with the smallest $D_h$ and $D_v$: 117.0 mm and 89.3 mm. Because the aspect ratio of the display used in the experiments was 16:9, the horizontal gaze movements tended to be longer than the vertical movements; therefore, a larger $D_h$ than $D_v$ would be needed to prevent an unintentional detection.

#### 6.5.2 Analysis of Gaze Trajectories

We observed that, among the 26 unintentional dwell-then-gestures, some of the two-level strokes took too much time. To examine this, we calculated the time taken for the two-level strokes (hereinafter, the stroke time), i.e., the duration from the time the gaze started to move to the time the two-level stroke was detected. Figure 9 shows the frequencies of the stroke times for both intentional manipulations (obtained during the task using gesture-driven application) and unintentional manipulations (obtained during the task using a dwell-driven application). The average stroke time of the intentional dwell-then-gestures was 362 ms (SD = 137). Specifically, the stroke times of most two-level strokes ranged from 100 to 800 ms (Figure 9, upper). By contrast, some of the two-level strokes took a relatively longer amount of time, which should be unintentional dwell-then-gestures (Figure 9, bottom). This suggests that we can prevent unintentional detections by limiting the range of the stroke times. For instance, by limiting the range of the stroke times to the average ± one SD (225–499 ms), two SD (88–636 ms), or three SD (0–773 ms), 81.2%, 46.9%, and 25.0% of unintentional detections can be eliminated. Moreover, adopting 773 ms as the limit can eliminate 25.0% of unintentional detections while enabling the detection of intentional dwell-then-gestures, as shown in Figure 9, upper.

#### 6.5.3 Summary of Analyses

The following are the findings of the above analyses and the results of Experiments 1 and 2, which we used to fine-tune the parameters:

- The large $T_{dwell}$ results in a reduction in unintentional detec-

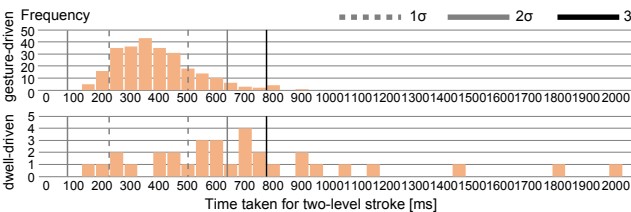

Figure 9: Frequencies of stroke times. The top graph shows the frequency of stroke times in the intentional manipulations. The bottom graph shows the frequency of stroke times the in unintentional manipulations.

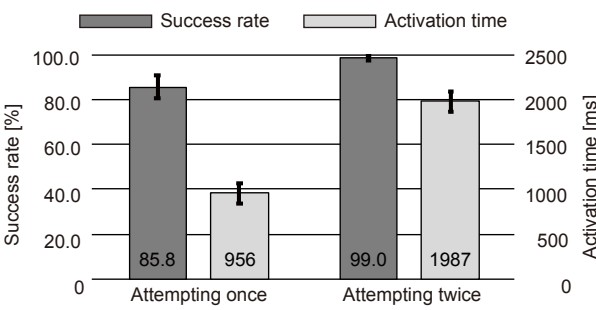

Figure 10: Success rate and activation time with fine-tuned parameters in the first (left) and second (right) attempts of the participants.

tions and an increased success rate. However, we can not use the larger $T_{dwell}$ as a fundamental solution to prevent an unintentional detection.

- A small $W_{path}$ may have trouble in preventing unintentional detections. To maintain a high success rate, $W_{path}$ should not be small.
- Horizontal gaze movements seem to be longer than vertical gaze movements. Therefore, $D_h$ should be larger than $D_v$ to keep the success rate high.
- We can regard a two-level stroke taking more than 773 ms as an unintentional detection.

Moreover, a proper combination of these findings (i.e., a proper combination of parameters) will lead to both zero unintentional detections and a high success rate (i.e., easiness of the command activation).

## 7 Fine-tuned Parameters

We used the above findings to fine-tune the parameters using the data from Experiment 3 and derived the following: a $T_{dwell}$ of 506 ms, $W_{path}$ of 34.6 mm, $D_h$ og 116.0 mm, and $D_v$ of 66.9 mm. These parameters achieved the highest true-positive rate (40.6% of success rate in the simulation) and zero unintentional manipulations. Interestingly, 116 mm was 21.8% of the horizontal length of the display (531 mm), and 66.9 mm was 22.3% of the vertical length of the display (299 mm). Although the absolute distance of a gaze movement varies depending on the display, the relative distance seems to be approximately constant (i.e., approximately 22.1%) between the horizontal and vertical movements. That is, users can activate a command by moving their gaze relatively the same distance on the display regardless of whether it is in the horizontal or vertical direction. Separating $D_{thld}$ into $D_h$ and $D_v$ may also be reasonable because humans have a larger horizontal field of view compared with their vertical field of view (left, 90°; right, 90°; upper, 50°; and bottom, 80°) [26]; in addition, this result is consistent with the results by Møllenbach et al. [23], where performing a single gaze gesture (one-level stroke) in the vertical direction takes longer than that in the horizontal direction. In addition, we can limit the stroke time to 0–773 ms; even when a gaze movement satisfies the parameters, the stroke is not detected as a two-level stroke when its stroke time exceeds 773 ms.

## 8 Success Rate with Fine-tuned Parameters

We conducted a study to investigate the success rate when the fine-tuned parameters are used. Ten volunteers (all male, including students in our laboratory) aged 21–25 participated. The same system as in Experiment 2 was used; the task was to perform a dwell-then-gesture from one target (C) among eight gestures. We asked to the participants to perform a dwell-then-gesture until it was correctly detected. The participants calibrated the eye tracker at the beginning of the study. They then practiced dwell-then-gestures in each direction for at least five minutes and conducted one practice session. After the practice session, they conducted five test sessions with

one minute rest between each session. In total, we collected data on 400 dwell-then-gestures (8 gestures × 5 sessions × 10 participants) starting from the center of the display. The experiment took approximately 25 minutes per participant.

The success rate (Figure 10, gray graphs) averaged for all gestures and over all participants was 85.8% (SD = 5.1) when the participants attempted to perform a dwell-then-gesture once. The highest rate was 90.0%, and the lowest rate was 72.5% across all participants. The average success rate when the participants attempted to perform a dwell-then-gesture twice was 99.0% (Figure 10, light gray graphs). Surprisingly, a participant with a rate of 72.5% for the first attempts had a rate of 97.5% for the second attempt. These results suggest that even when users of our system fail to activate a command, they can attempt to do so again without being affected by an unintentional manipulation. Moreover, the average activation time was 1,987 ms when attempting to perform a dwell-then-gesture.

Regarding the range of stroke times (0–773 ms), only one two-level stroke was not detected because the stroke time exceeded the range. The average stroke time was 450 mm in the first attempt and 525 mm in the second attempt; this difference between the two times may have been because the participants more carefully performed the two-level stroke during the second attempt than during the first attempt. Note that the range of the stroke time was derived in Experiment 3, where the parameters differed from those used in this study. However, the range worked well in this study. Therefore, the range could also be used with other combinations of parameters.

## 9 Discussions

### 9.1 Effects of User Attributes and Experimental Conditions

One participant in Experiment 1 suffered from nystagmus (rapid involuntary movements of the eyes). We found that the gaze trajectories of some of the eliminated trials were dispersed and that these trials were performed by this participant. In Experiments 2 and 3, none of the participants had eye disorders. Moreover, the experiments were conducted with only young participants in controlled environments with displays of the same size. To fine-tune the parameters further, we need to conduct more experiments to obtain gaze trajectories under various environments and more varied situations, such as while users watches a movie, applying user attributes to ensure that our technique is suitable as a general command activation technique. In addition, collecting 24 hours of gaze data in a real-world situation is necessary i.e., all gaze movements caused in daily life (such an investigation was conducted in a smartphone-based interaction design e.g., [1]).

### 9.2 Effects of Distractions

During the experiments, no objects unrelated to the experiment were displayed. However, under normal circumstances, a variety

of GUIs and text may be displayed on a screen; such distractions may have certain effects. Two participants of Experiments 2 and three participants of Experiment 3 commented that performing the two-level strokes was difficult because there was no object that could be referred to; in addition, another participant commented, "using the edge of the display, I feel that performing the two-level stroke was relatively easy". These findings suggest that distractions might positively influence the results of our technique. However, distractions may also have negative effects. For example, when the users perform a two-level stroke, their gaze could be targeted toward a blinking object and thus move outside the path of the stroke. In the future, we should therefore investigate the effects of distractions.

### 9.3 Using a One-level Stroke

Using a one-level stroke as a command activation technique will also be attractive. However, to do so, we should limit the parameters more restrictively than those used for detecting a two-level stroke. We ran another simulation to find the combinations of parameters that can detect one-level strokes with the gaze trajectories obtained in Experiment 3. As a result, there were numerous combinations of $W_{path}$ and $D_{thld}$ for which no unintentional one-level strokes were detected. For example, if we use a $T_{dwell}$ of 506 ms, the combinations would have a small $W_{path}$ (e.g., less than 5.3 mm), a large $D_h$ (more than 368 mm, which is 69.3% of the horizontal length of the display), and a large $D_v$ (more than 167 mm, which is 55.9% of the vertical length of the display). With these parameters, the gaze movement distance becomes substantially long; for example, users need to gaze from the left edge of the display to the right edge of the display. In this design, users may be able to activate a command more easily than when using a two-level stroke; however, the start position of a one level-stroke must be near the edge of the display. Therefore, one-level strokes may be useful gestures for command activation on the entire window, and further investigations should be conducted in this regard.

### 9.4 Using Other Gesture Detection Algorithms

Our system works in the same way as other gesture recognition algorithms such as $1 Recognizer [33]; that is, the system inputs a trajectory and outputs a recognition result. In fact, we tried to use $1 Recognizer for gesture recognition. However, detecting start and end points remains a challenge; if the detection is performed everywhere on the display, it might cause an unintentional recognition because the system will output the recognition results continuously. To address this challenge, the gesture detection system of Rajanna and Hammond [25] uses a modified version of $1 Recognizer. The point at which the gaze starts to move is detected when the user fixates at the top-left corner of the display. Although this method seems to solve this problem, it appears to be disadvantageous because object selection cannot be supported. By contrast, our gesture detection system can find a start point and an end point of the gaze movement. Moreover, the trajectory between the start point and end point can be applied to these gesture detection algorithms to recognize the shape of the trajectory, which will further expand the design space of a gaze-based interaction. In addition, comparing our technique with these algorithms will be necessary.

### 9.5 Advanced Interactions

Incorporating more interactions would improve our technique.

#### 9.5.1 Applying Marking Menu Principles

Our technique does not provide users with visual guidance; thus, novices may be confused with mapping between the commands and two-level strokes. To solve this problem, we plan to use a marking menu [18, 19] that can support both novices and experts. Namely, novices will be able to activate a command by referring to the displayed menu if they have any difficulties. By contrast, experts can simply activate the command, i.e., they do not need to activate the menu. At present, our technique is only suitable for experts because of the short time dwell (506 ms) and lack of visual guidance. To provide support to novice users, we plan to use a long dwell time, e.g., 1.3 s (the zero dwell detection time in Figure 7), to display visual guidance. However, we should carefully determine the timing of the command activation because there are two types of gaze movements: one checks a mapping of a command to a gesture and the other is the gesture itself. One idea for determining the timing is to use the gaze data collected while referring to a cheat sheet in Experiment 2 because the data consist of the gaze movement for checking the mapping and time taken for checking the mapping. Moreover, in [9], the necessity of a delay for displaying the menu is investigated and the design guidelines regarding to the delay are shown. Supporting these guideline for gaze-based interaction using our technique will contribute to the field of gesture-based manipulation.

#### 9.5.2 Specialized User Interface

The results of Experiment 2 showed that our gesture detection system cannot be used at the corners of the display. The specialized user interface of our technique might be a way to deal with this problem. One solution is to assign at most two commands at a corner of a display (e.g., mapping two commands to each $R{\rightarrow}D$ and $D{\rightarrow}R$ when the position is the top-left corner of the display) or four commands at the center of the edge of a display (e.g., mapping four commands to each $R{\rightarrow}D$, $D{\rightarrow}R$, $R{\rightarrow}U$, and $U{\rightarrow}R$ when the position is the center of the left edge of a display). This might work because the success rate of a gesture in which the gaze moves along the edges of the display (e.g., when the position is TL and the gesture is $R{\rightarrow}D$) tended to be high in Experiment 2.

#### 9.5.3 Dynamic Parameter Calibration

Calibrating the parameters to the characteristics of each user would make our technique more robust against unintentional manipulation; the calibration would make $W_{path}$ smaller, making $D_h$ and $D_v$ larger, and narrow the range of the stroke times. Moreover, to ensure that the command activation is easy to apply under various situations, the calibration should be based on gaze trajectories dynamically obtained from intentional command activations of the user. By developing such a dynamic calibration system, the parameters can be fine-tuned for various situations, i.e., not only for the applications used in the experiments.

## 10 Conclusion

We showed a gaze-based command activation technique that prevents unintentional manipulations. Using our technique, users can activate a command through a series of dwell-then-gesture manipulations. The gesture adopted is a simple two-level stroke, which consists of a sequence of two orthogonal strokes. Our technique allows users to activate a command on an object such as 'opening' a file and a command on a window (i.e., not on a specific object) such as 'switching' the foreground window. Although this scheme works well for mouse-based manipulation, in a gaze-based interaction, it has been unsuccessful owing to the problem of unintentional manipulations. To achieve robustness against unintentional command activations and the same scheme using a mouse-based manipulation, we designed and fine-tuned a system for detecting a user's intentional dwell-then-gesture manipulation based on how the users move their gaze, as revealed through three experiments. The detection system was designed mainly for preventing an unintentional command activation, and its success rate of intentional command activation was 85.8%. However, users can activate a command with a 99.0% success rate if they perform a dwell-then-gesture twice; this means that users do not need to worry about unintentional command activations because of the technique's robustness against an unintentional

command activation. Although our technique seems to simply combine well-known dwell- and gesture-based manipulations, implying a low rate of success, our approach is the first gaze-based command activation technique that consists of a series of a short time dwelling for target selection and a simple gesture for command activation, and will become the first technique adopting a marking menu, which is a traditional menu used for command activation in such as mouse- or pen-based interactions to gaze-based interaction.

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
