# OpenReview forum: "Gaze-based Command Activation Technique Robust Against Unintentional Activation using Dwell-then-Gesture"
_graphicsinterface.org/Graphics_Interface/2020/Conference — GI 2020_

### Official Review · AnonReviewer1 · 2020-04-08
**Fairly dense and presentation not perfect, but an interesting idea worthy of publication**

**Rating:** 7
**Confidence:** 4

**Review:**


This submission explores the question of how to design gaze-based interaction techniques. The authors propose and refine a gaze-based dwell+gesture technique through several micro-studies and simulation. They demonstrate that the technique is effective for what seems to be an acceptable amount of time without too many erroneous activations through a final study (which curiously is not described as an experiment). The contribution here is a technique that ameliorates accidental dwell activations through a more extensive activation step.

I recommend this manuscript be accepted.

While I am not an expert in the gaze-based interaction space, it seems here that there is something interesting that will be of benefit to the community. If nothing else, it contributes to the discussion of how to design these interaction techniques, and provides an example of a fairly extensive refinement process that helps the authors arrive at the well-defined model at the end of the manuscript.

I do think that there are some significant presentation problems in the manuscript that we see submitted here. At some points, the presentation problems interfere with the clarity of the ideas, and this is problematic. I have provided several opportunities for how I think the manuscript's clarity can be improved by approaching the presentation slightly differently.


# Abstract
* Would be better to clarify what these three experiments are and/or what was revealed about eye movement in these experiments.
* Do you evaluate the technique that you design here?
* It doesn't look like the last clause in the last sentence is written correctly.

# Introduction
* First two paragraphs are very well written. The third one is a bit confusing.
* The fourth paragraph begins in a very confusing way: "combination of dwell-based manipulation and gesture- based manipulation shows potential for gaze-based command activation", since the previous sentence says: "Although this scheme works in mouse-based manipulation well, in gaze- based interaction, it does not because gaze-based interaction still faces the problem of unintentional manipulations." These two seem to contradict one another.
* I feel it would be useful in the introduction to describe the experiments (why were they done? what do we learn? how does it motivate the design of your technique?)

# Experiment 1
* It would be useful to clarify the purpose of this study in the lede of the study description. Is this study to assess what is easy to do? Or is it to train a gesture recognition system?
* Where are the instructions (e.g. UR) displayed? Does a participant receive the instructions verbally from the experimenter before staring at the black circle? Or are the instructions illustrated on the black circle somehow?
* Analysis: It seems surprising that so many trials have these errors. Are these slips, or mistakes, or miscalibrations of the equipment? etc.
	* In Figure 2, this seems to represent UR. If the participant had gone slightly downward first (in the trial illustrated, they go slightly upward first), would the trial have been discarded? Reading the analysis description, it looks like it would have.
* I am a little confused by the description of what happens next, but it looks like overall, you are trying to develop a model that has parameters that are tunable that can account for slight variations in the trajectories as people travel from one point to the end of a stroke. If this is the case, then this needs to be clarified as the goal of the experiment in the lede.
* One thing that isn't clear is what happened to the single- and two-level stroke data. Were they treated the same? Were they separated in the analysis?

# Gaze Detection System
* This might be clearer if D_thld was renamed to W_dwell -- i.d. the width of the dwell point (since there is a large threshold before it is considered not to be a "dwell" anymore)

# Experiment 2
* Lede is much clearer
* I'm really puzzled why the design has participants having to do mapping between a letter and a gesture path. Wouldn't it be more prudent to simply have them look at an arrow that tells them which gesture to perform? (this would negate the necessity of a cheat sheet) It's also unclear to me why the study design provides the detection result -- the participants are simply creating data; it's not necessary for them to know how the system performed, right?
* Why does the experimenter do this? What is the purpose: "In description phase of the practice sessions, an experimenter told the participants their results and the reasons for their fail- ures."
* I *think* what is happening is that there are a series of trial rounds where the participant is "learning" how to do the gestures well, and getting feedback from the system and the experimenter. In the subsequent "train the computer" rounds, there is no feedback given, and THIS is treated as real data for a later phase. If this is the case, then this needs to be explained a little more clearly.
* The basic idea of simulation I understand. In this case though, is the idea to use some of the earlier data collected in the simulation? Would this be appropriate? It cannot properly simulate how a person might react, because if the system provides feedback of an early activation, the person may behave differently than they do for the collected simulation data (since this was collect without visual feedback)

# Experiment 3
I understand the basics of this

# Conclusion
* Be careful of the final claims you make here. You *do not* make it equal to mouse-based interaction.

---

### Official Review · AnonReviewer2 · 2020-04-16
**Interesting technical work but not well positioned in the literature and with questionable validation**

**Rating:** 4
**Confidence:** 4

**Review:**

This submission mostly reports on the technical description of a gesture recognition algorithm for non-guided gaze-based command activation in gaze-only controlled interfaces. It describes four main studies.

The first one characterizes one- and two-level gaze gestures when performed without guidance and from a point located at the center of the display. The trajectories collected in this study are then used to design a gaze-gesture recognition system for detecting two-level gaze gestures, and to set the initial values of all its parameters.

The second study attempts to fine tune the parameters of the gaze-gesture recognition system for two-level gaze gestures performed from different locations on screen. In that respect, participants were instructed to perform a gaze-gesture from one of 5 different locations on screen, and were then notified wether the correct gesture was recognize.

The third study is focused on fine-tuning the parameters of the gesture recognition algorithm in order to optimize its performance regarding the correct recognition/accidental activation trade-off.

Finally, a fourth study investigates correct command selection rate with fine-tuned parameters.

This is a very dense paper, with lots of content and reporting (to varying level of details) four user studies.

The scope of the paper, that is the design of a gaze-gesture recognition algorithm is interesting. However, the work suffers in my opinion from the two following main limitations.

#Scope and positioning
Gaze-gesture has been studied for years in the accessibility community, and also explored more recently in the context of head-mounted displays as an alternate method notably for command selection. However, the paper quickly dismiss previous gaze-based gesture systems to quickly conclude that non-guided gaze-based selection will be explored, and jump into building a gesture recognition algorithm without carefully explaining why existing algorithms are limited and would not work.

More precisely, several gaze-gesture methods have been explored, from unidirectional gestures [19,20] to pie menus [24,A] to smooth-pursuit based gestures [3]. While the current submission cite these works, it right away decides to investigate the very specific instance of *non-guided* *two-level* gaze gestures not relying on smooth-pursuit, and to investigate in depth a recognition algorithm for this specific case without providing any rationale abut how this design is sound. Guidance is a critical component of both command selection [B] and gesture execution [C] and dismissing these aspects for a gaze-based command selection mechanisms would require to carefully discuss these aspects (earlier than a paragraph in the discussion) and possibly to investigate their impact on user performance using the proposed system. Moreover, command selection is a task that involves several component that need to be discussed (see tables in [B]). Otherwise, if users cannot browse the commands, it is mostly a shortcut mechanism.

Similarly, gesture recognition can be achieved in many ways, and different methods have been employed to recognize gestures, both in general and in gaze-based interfaces  (1$ recognizer, DTW, Rubine, classification using SVM, [E], Knn, etc.). Current submission barely discusses why these methods cannot be applied directly. One specificity of gesture recognitionin gaze-only controlled interfaces is that the gestures must be self-delimited thus requiring most of existing algorithm to be adapted to this context. However, DTW and Knn have already been demonstrated as efficient in the context of self-delimited 3D gesture recognition [E,F]. Unfortunately, current submission does not carefully explain why existing gesture recognition method are not investigated, and the proposed solution is not compared to any baseline either.

Therefore, the submission falls between two stools. It does not convincingly present or validate a novel gaze-based command selection technique; it does not convincingly present or validate a novel gesture recognition algorithm for gaze-based command selection.

Instead, it swiftly starts to investigate the design and fine-tuning of an ad-hoc gesture recognition algorithm, which is fine, but need to be more carefully motivated and validated.

#Validation of the proposed technique
The four studies are interesting, but several aspects are highly questionable.

First, it could be anticipated (notably because it is raised by authors in studies 2 and 3) that on-screen location as well as the objects displayed on screen can influence the accuracy of gaze gestures. Therefore I was surprised that study 1 characterizes gestures from the center of the screen only, leaving other locations to study 2 (with different goals and procedure). A more generic characterization of gaze gestures should probably use location and displayed content as factors from the start. Instead users iterate on this characterization (or more precisely, on the inferred parameters) along the studies.

Then, study 3 follows a surprising and unconventional experimental procedure to investigate the robustness of the algorithm against unintentional activations. Indeed, it asked users to perform reading and text entry tasks in which dwells were used specifically for activating commands. Previous work investigating the Midas-touch problem [E,F,G] tests accidental/unintentional activation by collecting data (in this case, eye-tracker information) while using the device in real-world situation for a given duration (e.g. 24h in E). In contrast, authors decided to collect data from artificial tasks while users interacted using a ad-hoc dwell-based command selection technique. As a result, it is unclear how collected data is valid to test for unintentional activation.

Finally, and combing back to previous step, the proposed method is not compared against other gaze-gesture command selection mechanisms

# Main recommendations/suggestion
That being said, the work has still value, but the lack of careful positioning and motivation, combined with questionable validation makes me reluctant to accept this paper. My main suggestion for alleviating this problem would be as follow

First, carefully describe the context, motivation and research questions. Describing an algorithm for gaze-based command selection requires to 1-carefully explain why the chosen command-selection mechanism is sound (why two levels gestures, why no visual aid, why no smooth pursuit such as in G3 [3]) and why existing gesture recognition algorithm cannot work or be adapted. So far, this aspect is briefly summarized at the end of  the related work as "In our technique, a user can activate a command by performing a dwell and then a simple two-level stroke." but does not explain why it would be more efficient than G3 [3] for instance.

Second, assuming that the work carefully motivates non-guided gaze-based command activation in gaze-only controlled interfaces, I would suggest to rearrange the experiment 1, "gesture detection system", experiment 2 and "fine-tuned parameters" sections. It currently uses an iterative design methodology without having the corresponding section. Gestures are characterized in experiment 1, then gesture detection system is explained, and then experiment 2 tests it in different conditions, which yields that parameter values are not adequate and uses a simulation to update their values, and so on. Rather, the submission should first more carefully and comprehensively characterize the gestures, and then describe a gesture-detection system. Fine tuning the parameters afterwards is sound and makes sense, but iteratively doing it after each experiment using simulation makes it harder to follow.

Third, compare the performance of the proposed gesture-recongition system both in term of true and false positive, with a baseline from the literature.

[A]- https://doi.org/10.1145/1344471.1344483
[B]- https://doi.org/10.1145/3002171
[C]- Gordon Kurtenbach. PhD Thesis. The design andevaluation of marking menus.University of Toronto. 1993
[D]- https://www.autodeskresearch.com/publications/scaleindependencemm
[E]- https://doi.org/10.1145/1866218.1866265
[F]- https://doi.org/10.1145/2070481.2070503
[G]- http://doi.org/10.1145/2807442.2807489



=======================================================
Minor comments

- p1, intro: [3] uses Dwell but as far as I remember, I believe it is not literally gaze dwell, but a cursor that is at the center of the Head-mounted display
- p2, RW: [27] tests two different pursuits technique. Therefore, saying that dwell "performed as well or better than THE pursuit technique" is misleading
- p2, RW: there might be a sentence or transition missing before the last paragraph of the "Dwell-based command activation" subsection
- p2, RW: why characterizing gestures in "one-level" or "two-level" strokes? maybe saying that the more complex the gesture, the less likely it is to result in false positive would be sufficient?
- p2, RW: I would cite [3] regarding guiding gesture, even though smooth-pursuit is not exactly gesture
- p2, Xp1: why using a chin rest if the idea is to characterize gesture execution, it does not sound extremely sound if the idea is to implement a system that could be used in real world.
- p3, Xp1: for reproductive research, how were directions counter-balanced?
- p3, Xp1: I find 9.8% of outliers really high. I would suggest a stronger justification for removing so much data
- p4, GDS: "In our technique, a dwell was detected when the gaze stays in 5 mm (t1) for the dwell time (Tdwell).". Why using 5mm if this is the accuracy of the Tobii? (that is an offset of 5mm can be measured). Wouldn't it be better to make it more permissive?
- p5, Xp2: Was there any interaction position x gesture direction interaction effect? No effect of group at this stage?
-  p7, Xp3: where the 26 unintentional dwell-then-gestures gesture recognized after a dwell on a button?
-  p7, Xp3: why simulating by changing parameters, while authors acknowledged that users may "learn" the technique. Therefore, simulation would miss this aspect.
-  p7, Xp3: I did not understand how longer dwell can result in a higher unintentional detection rate.
-  p9, Discussion: "To fine-tune parameters more, we need to conduct more experiments to obtain gaze trajectories in various environments and more varied situations," and "In the experiments, no object unrelated to the experiment was displayed."; Yes and Yes; In my opinion, these points are critical to characterize gesture and test the robustness of the algorithm
-  p10, Discussion: "Using Other Gesture Detection Algorithms"; Yes, and this should actually frame the contribution.
-  p10: Why increasing even more the delay to display the menu? Given that users have to dwell regardless to activate a command, length of the gesture is used for recognition, and participants complained about the lack of visual landmark guiding the gesture, then why not displaying the menu in order to provide guidance and prevent error from poor recall? see [Henderson et al. Investigating The Necessity Of Delay In Marking Menu Invocation. CHI 2020]

---

### Official Review · AnonReviewer3 · 2020-04-21
**Relatively solid, minor but novel contribution.**

**Rating:** 6
**Confidence:** 4

**Review:**

This paper presents the results of studies that examined using dwell and stroke gaze-based gestures. Through a series of experiments, the paper presents an analysis of how well dwell-then-gesture-based gaze interactions can perform.

Overall, the paper is well written. It also makes progress in an emerging area that is especially important to accessibility applications.

Regarding drawbacks:
#1 There are some language and phrasing issues but I think those are fixable.
#2 There are some issues with the study, regarding possible confounds due to lack of counterbalancing, but these may be minor.
#3 Lastly, and as the Abstract acknowledges, this can be seen as a relatively minor improvement and contribution.


#1 - Language
While the majority of the paper is clear and easy to read, there are some sections that are difficult to understand which impacts the readability and comprehension of the contribution.  For instance, what does it mean to 'enrich the vocabulary', and what does 'as much as mouse-based interaction'. I cannot understand this, and it is supposed to be the succinct definition of the contribution.


#2 - Study Execution and Reporting
In Experiment 3, which is potentially the most interesting, the task order was not counterbalanced. There may be a good reason for this, but it should be better detailed in the paper. Additionally, can more support be added as to why the particular study design and order of tasks was chosen?

In statistical reporting, please include the actual p-value, as well as the test statistic. For example on page 4, a 'Wilcoxon signed-rank test showed no significant difference (p < 0.05)' - this sentence is confusing because p < 0.05 would be considered significant by most, so its unclear if the wrong test was done, or the wording is confusing.

#3 - Contribution
The paper acknowledges that this can be seen as a combination of prior works. I think that is true to some degree. I think the contribution is a bit minor - I'm concerned that such large, eye-fatiguing gestures were used. I'm concerned that such precise parameters are tested and reported on. I'm concerned that the results will not be generalizable.

To me, the paper looks, for the most part, to be methodologically sound, complete in its reporting, and if some changes can be made, I do believe it will contribute to our global body of knowledge around eye gestures with the data it presents.

---

### Meta-Review · Area_Chair1 · 2020-04-21

**Recommendation:** Accept
**Confidence:** 4

**Metareview:**


Based on the reviews, I recommend that this manuscript be accepted to Graphics Interface.

While there are some issues with the presentation, both R1 and R3 are confident that there is a small, interesting contribution that is worthwhile publishing for the benefit of the community.

All reviewers noted several points where the authors can clarify their motiviation, their contribution, and relationship to prior work. R2 and R1 provides several detail spots where this can be done, and I want to draw the authors' attention to R2's appeal to provide a slightly more thorough treatment of prior work--specifically to clarify the contribution of this work, and the motivation and justification for the approach given prior work.

---

### Decision · Program_Chairs · 2020-04-25

Accept